# Spontaneous Learning of Visual Structures in Domestic Chicks

**DOI:** 10.3390/ani8080135

**Published:** 2018-08-06

**Authors:** Orsola Rosa-Salva, József Fiser, Elisabetta Versace, Carola Dolci, Sarah Chehaimi, Chiara Santolin, Giorgio Vallortigara

**Affiliations:** 1Center for Mind/Brain Sciences, University of Trento, 38068 Rovereto, Italy; orsola.rosasalva@unitn.it (O.R.-S.); carola.dolci93@gmail.com (C.D.); saretth@libero.it (S.C.); 2Department of Cognitive Science, Central European University, 1051 Budapest, Hungary; fiserj@ceu.edu; 3School of Biological and Chemical Sciences, Queen Mary University of London, London E1 4NS, UK; e.versace@qmul.ac.uk; 4Centre for Brain and Cognition, University Pompeu Fabra, 08005 Barcelona, Spain; chiara.santolin@upf.edu

**Keywords:** domestic chicks, *Gallus gallus*, imprinting, implicit learning, statistical learning, sequence learning, spatial/visual configurations, positional information

## Abstract

**Simple Summary:**

Our aim is to investigate the recognition of the structure of multi-element configurations; one mechanism that supports communicative functions in different species. Cognitive mechanisms involved in this ability might not have evolved specifically for communicative use, but derive from other functions. Thus, it is crucial to study these abilities in species that are not vocal learners and with stimuli from other modalities. We know already that domestic chicks can learn the temporal statistical structure of sequences of visual shapes, however their abilities to encode the spatial structure of visual patterns (configurations composed of multiple visual elements presented simultaneously side-by-side) is much less known. Using filial imprinting learning, we showed that chicks spontaneously recognize the structure of their imprinting stimulus, preferring it to one composed of the same elements in different configurations. Moreover, we found that in their affiliative responses chicks give priority to information located at the stimulus edges, a phenomenon that was so far observed only with temporal sequences. This first evidence of a spontaneous edge bias with spatial stimuli further stresses the importance of studying similarities and differences between the processing of linguistic and nonlinguistic stimuli and of stimuli presented in various sensory modalities.

**Abstract:**

Effective communication crucially depends on the ability to produce and recognize structured signals, as apparent in language and birdsong. Although it is not clear to what extent similar syntactic-like abilities can be identified in other animals, recently we reported that domestic chicks can learn abstract visual patterns and the statistical structure defined by a temporal sequence of visual shapes. However, little is known about chicks’ ability to process spatial/positional information from visual configurations. Here, we used filial imprinting as an unsupervised learning mechanism to study spontaneous encoding of the structure of a configuration of different shapes. After being exposed to a triplet of shapes (ABC or CAB), chicks could discriminate those triplets from a permutation of the same shapes in different order (CAB or ABC), revealing a sensitivity to the spatial arrangement of the elements. When tested with a fragment taken from the imprinting triplet that followed the familiar adjacency-relationships (AB or BC) vs. one in which the shapes maintained their position with respect to the stimulus edges (AC), chicks revealed a preference for the configuration with familiar edge elements, showing an edge bias previously found only with temporal sequences.

## 1. Introduction

The ability to detect regularities in the sensory input is crucial for communication. For instance, human language and birdsong require the processing of complex structures of auditory stimuli [1,2,3]. However, some cognitive abilities underlying these communicative adaptations might not have evolved *de novo* to support them. Rather, they might have been co-opted from other cognitive mechanisms used for visual processing or for learning action sequences, including “statistical learning” (general mechanisms that enable the acquisition of structured information and the detection of regularities in the sensory input [1,2,3], reviewed in several previous papers [4,5,6]). 

To understand the origins of communicative systems, we must investigate these mechanisms from a comparative perspective in nonvocal learning animals and nonlinguistic tasks (e.g., with visual stimuli structured in space [4]). This allows to identify mechanisms that are common to various sensory modalities and species and others that are not [6,7]). Here we follow this approach and study implicit learning of spatially defined visual sequences in young domestic chicks.

Statistical learning of temporal sequences has been implicated in birdsong and in human language (reviewed previously [6]). The first step to process linguistic input is to parse the continuous speech stream into words [8]. This can be done by recognizing the frequency with which groups of sounds occur in a given order, an ability shown by human infants and adults [9,10,11,12]. However, this ability is not restricted to linguistic or even auditory input [13,14,15] and is shown also by nonhuman animals [3,16,17,18,19,20]. Likewise, other abilities necessary to the evolution and acquisition of linguistic syntax, investigated by studies that test humans’ and animals’ capability to recognize various kinds of regularities in the temporal order of a sequence of elements [21,22,23,24,25,26,27], are not restricted to language [28,29,30] or to humans [23,25,31,32,33,34,35,36,37,38,39,40,41,42,43]. However, the use of linguistic inputs and communicative signals facilitates some of these tasks for human infants [44,45].

These abilities seem to be supported by general mechanisms that can process different types of stimuli, such as auditory linguistic stimuli, visual patterns, or touches (even though they are modulated and constrained differently in the way they operate across modalities and domains [4,6,46]). For example, visual statistical learning has also been investigated for the processing of spatial information (i.e., the spatial relationship between multiple visual elements simultaneously presented in different spatial positions). These studies showed that both human adults and infants spontaneously learn the properties of spatially defined visual patterns [15,47,48], which helps infants with the initial structuring of the visual environment [49]. Similar evidence has been reported also in adult animals, using conditioning procedures [50]. Spatial configurations of visual elements have also been used also to study the capacity to create abstract representations that can be applied to new exemplars, both in human infants [51] and in nonhuman animals [52,53,54,55,56,57].

Infants’ ability to recognize spatially defined series (linear arrangements) of visual elements was studied with a method similar to that used in the present work. Sequences of three audiovisual elements (conventionally labeled as A, B, and C) were organized both by their temporal and spatial order, according to the sequence ABC or CAB. Sensitivity to the spatiotemporal order of the elements has been demonstrated by showing, for example that infants habituated to the stimulus ABC and discriminated it from CAB [58,59].

To understand the development of linguistic and visual processing, it is important to study what kind of learning can develop in an unsupervised way, encoding the structure of complex inputs without direct feedback or reinforcement [60]. Chicks of precocial species [61,62] are particularly advantageous on this regard, thanks to the learning phenomenon of filial imprinting [63]: they learn, by mere exposure and without reinforcement, the features of the conspicuous objects they are exposed to, and they restrict their approach and affiliative responses to those objects (reviewed previously [64,65,66]). Although, in the chicks’ natural setting the imprinting stimulus will be a single object (the mother hen), there is increasing evidence that imprinting might apply also to a grouped configuration of elements [18,42,53,57].

In the current study, we used domestic chicks to take advantage of this powerful form of spontaneous learning. Domestic chicks present also other advantages to investigate the fundamental mechanisms at the basis of communicative functions, since they are the precocial offspring of nonvocal learners: the encoding of environmental regularities may show some differences between vocal learners and nonvocal learners and precocial and altricial species [6].

Based on these advantages, in a recent paper using filial imprinting we investigated chicks’ spontaneous capability to recognize the structure underlying a stream of visual stimuli, revealing a remarkable capacity to recognize the temporal order of pairs of shapes [18]. In contrast, little is known about chicks’ capacity to spontaneously encode the spatial relationship between multiple visual elements all simultaneously present in the visual scene, although this ability has been extensively investigated in human infants and adults [15,47,58].

Importantly, temporal and spatial information require somewhat different types of learning. While spatial configurations of visual elements reduce constraints due to working memory limitations, they also allow for the encoding of different structural properties than temporal sequences. For example, in the study of Santolin et al. [18], the only predictable features were those defining the ordered pairs (“shape A will always be followed by shape B”). In contrast, when simultaneously visible elements maintain fixed reciprocal spatial positions (such as in a previous paper [58]), subjects can potentially encode a higher variety of properties. For example, given the ABC sequence, an organism could potentially encode the stimulus structure in a multitude of ways. One could represent which shapes are adjacent to each other (A and B are adjacent, A and C are not), the distance dependencies between them (A will be followed by C, with an interleaving element separating them), the left-right order of the elements (A is the first element from the left, B the second etc.), the position of the elements in relation to reference points such as the stimulus edges (A and C are next to the stimulus outer edges, B is not), and so on. Thus, it is unclear if and how learning abilities observed for temporal sequences will translate in this context that offers richer structural information to encode. Previous studies showed the ability of chicks to learn the color configurations of their imprinting stimuli, abstracting the general pattern characterizing them [42], but the positional components of these abilities, and the role of shape configurations is unknown. The aim of the present paper is to make a first step in this direction, by investigating young chicks’ spontaneous learning of the structural properties of a configuration of visual shapes simultaneously presented in a fixed spatial order [58]. Although we used moving stimuli to attract the animals’ attention (imprinting is more effective with moving stimuli), all the elements of each stimulus were always simultaneously visible on the screen, therefore, the structure of each stimulus was defined by the relative spatial positions of the elements and not by their temporal order of appearance.

## 2. General Materials and Methods

### 2.1. Subjects and Rearing Conditions

Only female chicks (*Gallus gallus domesticus*) of the Aviagen ROSS 308 breed were used, since a pilot experiment showed that females but not males exhibit consistent preferences with these stimuli. It has been previously shown that the imprinting preference can be a sexually dimorphic trait in chicks, potentially masking recognition effects in one of the two genders [57,67]. Hence, to simplify the experimental design and reduce the number of tested subjects we decided to focus only on females. In this strain, sex can be determined using the wing feathers dimorphism.

Eggs were obtained by a local commercial hatchery (Agricola Berica, Montegalda, VI, Italy) and were kept in the darkness, inside a MG 140/200 Rural LCD EVO incubator (FIEM srl, Guanzate (Como), Italy) until day 18 of embryonic development. During this incubation stage the temperature was of 37.7 °C and humidity was 40–45%. On the 18th day of incubation, the eggs were moved to an MG 316H EVO hatchery (FIEM srl, Guanzate (Como), Italy), where they were still kept in darkness and at the same temperature, but with 60–70% humidity. Chicks hatched at the 21st day of incubation and were individually moved from the dark incubator to the housing facilities. Females were immediately housed in individual cages according to the experimental conditions. Males and all individuals after the test were housed in groups. Water and food were available ad libitum until chicks were donated to local farmers. All the animals were maintained at 29 °C, 68% humidity, under a natural 14:10 light:dark cycle.

Female chicks were housed individually with food and water available ad libitum in the imprinting cages, made of black plastic (30 × 38 × 33 cm, *w* × *l* × *h*) with a monitor (17′′, 60 Hz) mounted on the front wall (protected by a thin layer of plexiglass), that was used to play the imprinting stimulus (see below Section 2.2). During the imprinting phase, chicks could not see each other, since they were housed individually in black plastic cages that separated them visually from the other animals (this was done in order to prevent visual imprinting on conspecifics, which could have impaired learning about the artificial imprinting object).

### 2.2. Imprinting and Test Stimuli

During the imprinting phase, the monitor present in each cage played the imprinting stimulus continuously for 14 h each day (during the night, when the lights of the animal house were off, a black screen substituted the imprinting stimulus).

The imprinting stimulus was a configuration of three different red shapes, over a white background (see Figure 1). Each shape used to create the imprinting stimulus fitted in a 3.5 cm square (inter elements distances were of 0.65 cm). The stimulus was presented at a height of 3.2 cm from the lower margin of the screen.

For the entire imprinting phase, each animal was exposed to the same shapes presented in the same configuration (e.g., only “ABC” or only “CAB”).

Since movement elicits imprinting responses [68], the stimulus moved horizontally on the screen (covering approx. 24 cm), taking 10 s to run an entire cycle from right to left and back, in line with the procedure used in [57]. Thus, the absolute spatial position of each element on the screen varied.

During the test, chicks were faced with the choice between a familiar stimulus and a novel one (in Experiment (Exp.) 1), or between two stimuli that resembled the imprinting stimulus in different respects or to a different degree (Exp. 2 and 3). In all cases, both stimuli were composed of shapes taken from the familiar imprinting stimulus, but differed in the spatial arrangement of the shapes. For example, in Exp. 1 (Figure 1a), in the familiar stimulus the spatial ordering of the inner elements respected that of the imprinting object, whereas in the novel stimulus the familiar elements were presented in an unfamiliar spatial configuration. In Exp. 2 (Figure 1b), chicks were presented with fragments of the imprinting stimulus that either respected the between-elements adjacency-relationships of the imprinting stimulus (shapes that were close to each other in the imprinting stimulus were close to each other also in the fragment), or in which the elements maintain their position with respect to the stimulus edges (shapes that were at the edge of the imprinting stimulus were at the edge also in the fragment). Finally, in Exp. 3 (Figure 1c) chicks were faced with two test fragments, one of which respected the left-right orientation of the shapes in the imprinting stimulus, while the other violated it.

### 2.3. Test Procedure

On day 7 after hatching (after 6 days of imprinting), chicks were tested for their preference to walk towards each of the test stimuli. Before the test, chicks were food deprived for about 2 h to increase their arousal. Moreover, 20 min before the beginning of the test, an opaque black partition was used to occlude the screen in the rearing cage, preventing visual contact with the imprinting object until the moment of the test. This was done to increase motivation to approach stimuli resembling the imprinting object during the test, following the procedure used in [18,69].

For the test a running wheel (33 cm diameter, 12 cm wide) was used. The running wheel was suspended 2 cm above the floor at the center of a longitudinal runway (46 × 150 × 45 cm, *w* × *l* × *h*) whose interior surface was uniformly lined with black plastic. At the two ends of the runway, two video screens identical to those used for imprinting showed the test stimuli. The side of presentation of the two test stimuli was counterbalanced between subjects to rule out the effect of potential environmental asymmetries.

At the beginning of the test, chicks were individually placed in the center of the running wheel, facing one of the lateral walls, so that they could see both stimuli on the opposite sides of the apparatus. During the test, that lasted 20 min, the chick could walk towards either of the two stimuli. An automated counter measured the distance run (in cm) by the chick in each direction for the whole test duration. The test session was recorded by a video camera placed above the apparatus.

### 2.4. Data Analysis

For each chick, we analyzed the preference for the familiar stimulus (or for the stimulus with the familiar adjacency relationships in Exp. 2, and with the familiar left-right orientation in Exp. 3). This was expressed as a proportion of the distance walked towards the two kinds of stimuli, computed according to the formula:$$\mathrm{proportion}\;\mathrm{of}\;\mathrm{distance}\;\mathrm{run}=\;\frac{\mathrm{Distance}\;\mathrm{walked}\;\mathrm{towards}\;\mathrm{the}\;\mathrm{familiar}\;\mathrm{stimulus}}{\mathrm{Overall}\;\mathrm{distance}\;\mathrm{walked}\;\mathrm{in}\;\mathrm{both}\;\mathrm{directions}}.\;$$

Values of the proportion range from 0 (walking only towards the unfamiliar stimulus) to 1 (walking only towards the familiar stimulus). A value of 0.5 corresponds to the chance level. To verify if chicks discriminated between the two stimuli, values of the proportion of preference were compared to chance level by a one-sample two-tailed *t*-test. Wherever required, in order to compare experimental groups we ran independent samples *t*-tests. To further interpret nonsignificant results, in Exp. 3 we ran nonoverlapping hypotheses (NOH) Bayes factor (BF) analysis for the one-sample *t*-test case (scale r on the effect size = 0.707 [70]).

### 2.5. Ethical Statement

All applicable European and Italian guidelines for the care and use of animals were followed. All procedures performed were in accordance with the ethical standards of the University of Trento, where the study was conducted. The study has been approved by the research ethics committee of the University of Trento (OPBA) and by the Italian Ministry of Health (permit number 1138/2015 PR, 987/2017 PR).

## 3. Experiment 1

The aim of this experiment was to verify if chicks imprinted on a series of three shapes would spontaneously encode the spatial relationships between these elements, or whether this form of implicit learning by exposure, would result only in encoding of other less subtle properties of the imprinting stimulus (such as, for example, its color or the shapes of the individual elements).

In order to verify whether chicks can recognize the internal spatial relationships characterizing the familiar imprinting stimulus, we tested if chicks would show a preference for the familiar stimulus over an unfamiliar permutation of the same elements. This stimulus was composed of identical shapes as the familiar imprinting object, arranged in a different order.

### 3.1. Subjects

The sample of this experiment consisted of 64 female chicks.

### 3.2. Imprinting and Test Stimuli

Chicks were imprinted on either one of two series of three shapes (designated here by the letters A, B, and C, respectively). Both imprinting triplets were composed of the same three shapes and differed only by the order in which the shapes were arranged: ABC and CAB. Half of the chicks were imprinted on ABC, while the remaining subjects were imprinted on the triplet CAB (Figure 1a).

At test, all chicks were presented with the choice between these two triplets, one of which was the familiar imprinting object and the other could be differentiated from it only by the spatial arrangement of the elements (in this design, for part of the chicks, ABC is the familiar triplet and CAB represents the unfamiliar permutation of shapes, and vice versa for the rest of the chicks).

### 3.3. Results and Discussion

No difference was apparent (t_55.547_ = −0.708, *p* = 0.482; based on the Levene’s test, equal variances could not be assumed and the appropriate correction was applied) in the preference for the familiar triplet over its permutation between chicks imprinted on ABC (mean = 0.553, s.e.m. = 0.057) and those imprinted on the CAB triplet (mean = 0.603, s.e.m. = 0.040). The two groups were thus joined and the data from all the subjects were treated together for further analysis. In the overall sample, we observed a significant preference for approaching the familiar triplet over the unfamiliar permutation (mean = 0.578, s.e.m = 0.035, t_63_ = 2.247, *p* = 0.028, Figure 2).

This indicates that, after imprinting on a series of three shapes that maintain their reciprocal spatial positions, chicks can discriminate two triplets that differ only in the spatial arrangement of their composing elements. An identical preference for the familiar configuration was observed for chicks imprinted on ABC and on CAB. Thus, we can say that chicks not only can discriminate patterns based on the element order, but more specifically, that they encoded this property during the exposure phase and used it to recognize and preferentially approach the familiar imprinting stimulus at test.

This result sets the base for the subsequent experiments described in this paper, among other things, by providing us with clear expectations about the direction of the preference that chicks will reveal in this context, which is in this case a preference for familiarity, the most frequent result in female chicks [57,67] (see [18,42,71,72,73] for a theoretical discussion).

Overall, it is clear that in order to succeed in this test, chicks must have developed a mental representation of the spatial arrangement of the elements composing the familiar imprinting sequence. However, what is still unclear is which aspects of the familiar stimulus are encoded by this representation, since the two triplets, ABC and CAB, differ in many potentially relevant properties. The following experiments were thus devoted to verify which of these properties are spontaneously encoded by chicks during imprinting.

## 4. Experiment 2

With this experiment, we wanted to further investigate chicks’ capability to encode the spatial arrangement of the elements composing their imprinting stimulus. The discrimination between ABC and CAB, revealed by Exp. 1, could, in fact, be based on a number of different properties that allow the discrimination of the two triplets, any (or any combination) of which could have been used by chicks. For example, chicks imprinted on the ABC triplet could have encoded the adjacency-relationships between the triplet elements (A is adjacent to B, B is adjacent to C, A and C are not adjacent). Alternatively, chicks could have encoded the position of the elements in relation to reference points such as the stimulus margins, rather than relative to each other (e.g., A and C are located at the triplet edges, B is not). In this second experiment, we thus tested if chicks would recognize as more familiar a fragment taken from the imprinting triplet, that follows the between-elements adjacency-relationships of the familiar stimulus (e.g., AB), over a fragment that violates those adjacency-relationships, but in which the elements maintain their position with respect to the stimulus edges (e.g., AC).

### 4.1. Subjects

The sample of this experiment consisted of 60 female chicks.

### 4.2. Imprinting and Test Stimuli

Chicks were imprinted on the stimulus ABC, identical to the one used in the previous experiment. During the test, chicks were presented with the choice between two stimuli, each containing two of the elements of the imprinting triplet. One of those bigrams was a fragment taken either from the left or from the right side of the imprinting triplet (AB or BC, respectively). The other bigram was always AC, composed of the two edge elements of the imprinting stimulus. Half of the chicks were tested for their preference between AB (taken from the left side of the imprinting triplet) and AC, while the remaining chicks were tested with BC (right side of the imprinting stimulus) and AC (Figure 1b).

While AC violates the between-elements adjacency-relationships that define the imprinting stimulus, it resembles the imprinting stimulus in other ways. For example, as it is the case also for ABC, its left and right edges are marked by the elements A and C, respectively. On the contrary, the other two test bigrams, AB and BC, do not resemble the imprinting stimulus in this respect, since they both present B at one of their edges (the right and left edge respectively).

Differently from the previous experiment, here none of the test stimuli were perfectly identical to the imprinting triplet. In this case, the preference of the chicks was represented as the proportion of distance run towards the bigram that respects the between-elements adjacency-relationships (AB or BC), computed according to the same general formula described above.

### 4.3. Results and Discussion

No significant difference (t_58_ = −1.478, *p* = 0.145) appeared between the behavior of chicks tested with the AB vs. AC pair (mean = 0.386, s.e.m. = 0.048) and those tested for their preference between BC and AC (mean = 0.475, s.e.m. = 0.034). We thus joined these two groups for further analyses. Overall, chicks showed a significant preference for approaching the AC stimulus, as revealed by the fact that the average preference (proportion of the distance run) for the adjacency-relationships bigrams was below chance level (t_59_ = −2.281, *p* = 0.026; mean = 0.431, s.e.m. = 0.03, Figure 2).

This significant preference indicates discrimination of the two different kinds of bigrams by the chicks, revealing the presence of a flexible representation of the imprinting stimulus, which allows chicks to recognize some of its properties also when presented in fragments of a different length than the original imprinting object. More specifically, in this case, chicks favored the bigram that was consistent with the structure of the imprinting object with regards to the position of the elements in relation to the stimulus edges, over fragments that respected the between-elements adjacency-relationships present in the original imprinting pattern. In fact, only the stimulus AC contains at its own edges the two elements that are located at the edges of the imprinting triplet ABC. We can thus conclude that, when presented with a spatial configuration of simultaneously presented visual elements, chicks spontaneously encode the position of its elements in relation to the stimulus edges, rather than in relation to each other.

A tendency to prioritize encoding of the sequence edges over information embedded within it has already been reported in the literature for humans and other animals, although only for temporally defined sequences [7,35,43,74,75,76] (see the General Discussion). Here, for the first time, we found a similar effect with spatially defined series of simultaneously presented elements.

However, it is still unclear to what level of detail chicks encode the information on elements located at the edges of the stimulus. With the next experiment, we aimed to answer this question.

## 5. Experiment 3

Exp. 2 revealed that chicks spontaneously prioritize information about the items located at the stimulus edges over information about adjacency relationships between the elements. The aim of the current experiment was to further investigate the kind of information that is encoded by chicks concerning the elements that are located at the sequence edges. In particular, we wanted to verify whether chicks would encode only the identity of the elements that mark the two sequence edges or they could also differentiate between the left and the right sequence edge. To do so, we tested chicks’ choice between a bigram that resembled the imprinting triplet (ABC) in the left-right orientation of the two edge elements (AC) and another one, in which the two elements exchanged their respective left-right position (CA). 

### 5.1. Subjects

The sample of this experiment consisted of 76 female chicks.

### 5.2. Imprinting and Test Stimuli

As in the previous experiment, chicks were imprinted on the pattern ABC. During the test, chicks had to choose between approaching the bigram AC or the bigram CA (Figure 1c). Both the test stimuli contained, at their edges, the two elements that marked the edges of the familiar imprinting stimulus. The two stimuli differed only in in the left-right ordering of the edge elements, which respected the orientation of the imprinting stimulus for AC, but not for CA.

In this experiment, we thus computed the preference of the chicks for AC (again calculated as the proportion of distance run towards this stimulus), which was used as a dependent variable.

### 5.3. Results and Discussion

No significant preference was found for the fragment AC, characterized by the familiar left-right orientation, over CA (t_75_ = −1.592, *p* = 0.116, mean = 0.545, s.e.m. = 0.028, Figure 2). Even though the average level of preference seemed to suggest a trend in this direction, variability of the sample was too high, preventing this result from reaching statistical significance. We are thus unable to confirm that chicks spontaneously encode and/or recognize the left-right ordering of the edge elements of the imprinting triplet. Based on this result, we thus suggest that chicks could be sensitive to the fact that these elements should be specifically located at the edges of the sequence, without however discriminating between the left and the right edge. Nonsignificant results are notoriously of difficult interpretation and should be treated with caution. In order to strengthen our interpretation of these data, we tested this null result by Nonoverlapping Hypotheses (NOH) Bayes factors, obtaining a scaled JZS (Jeffrey-Zellner-Siow Prior) Bayes Factor = 2.374, which favors the null hypothesis indicating a more than two-fold higher chance that the null hypothesis is correct. Therefore, the data support the hypothesis that there is no difference between chicks’ observed performance in this experiment and the value of 0.5, expected in the absence of any preference between the two test stimuli. This could be in line with other evidence suggesting that the discrimination of stimuli which have been mirrored across the left-right axis is a particularly difficult task for animals, more so than the discrimination of stimuli which have been mirrored across the vertical axis [77,78,79,80,81,82].

## 6. General Discussion

The aim of this work was to investigate implicit learning of visual configurations, using filial imprinting to verify if domestic chicks spontaneously recognize structural properties of spatial multi-element arrays. Previous studies showed the ability of precocial bird species to learn structural features of imprinting objects, such as the presence of identic/different components [53,57]. Moreover sophisticated abilities had already been found in chicks for temporal visual sequences, revealing recognition of the order of appearance of shape pairs [18].

Similar to what had already been done with human infants [58], here we used patterns composed of three visual elements simultaneously presented in a fixed spatial configuration to investigate whether chicks recognized the structure of the imprinting stimulus. The stimuli used in the current study contain different structural properties, most of which were not available in the continuous temporal stream of shapes used by Santolin et al. [18]. Here we provide the first evidence of spontaneous encoding of spatial relationships between multiple elements all simultaneously present in the visual scene in this animal model. These results can open the way to further investigations of sensitivity to the structures of visuospatial displays in chicks, bridging the gap to the literature on human adults and infants [15,47,58]. The current paper confirms that imprinting learning can go beyond the perceptual features of a single object to those of a grouped configuration of elements [18,53,57], a necessary prerequisite for this kind of investigation. In addition, our results show that, although differences in shapes are less salient than color differences in imprinting [83], the shape can be taken into account to discriminate between stimuli (discrimination based on positional properties had been previously observed by a previous study [42] with manipulations of the color of imprinting stimuli).

More specifically, we showed, for the first time, that chicks spontaneously learn at least some information on the reciprocal positions of the visual elements since they discriminated stimuli such as ABC and CAB (Exp. 1). As the stimuli were constantly moving along the screen, chicks could not rely on the absolute position of items on the screen, but must have encoded relational spatial information to succeed in the task. Chicks were also capable of recognizing familiar properties of the imprinting stimulus from smaller fragments of it. In fact, after imprinting on ABC they discriminated AC from AB or BC (Exp. 2), indicating some degree of generalization. In this context, chicks’ preference for AC suggests that they prioritize information regarding the elements placed at the edges of the imprinting stimulus. However, chicks failed to discriminate between AC and CA, which differed only in the left-right orientation of edge elements (Exp. 3). Thus, chicks seemed not to distinguish between the left and the right margin of the stimulus. This could be due to the intrinsic difficulty of discriminating stimuli mirrored across the vertical axis [77,78,79,80,81,82], or to the specific bidirectional motion pattern that we employed in the present study. Our stimuli, in fact, constantly alternated rightward and leftward movements. This way, in the ABC triplet, the leading end was represented equally often by its rightmost or its leftmost element (A and C). Since young organisms seem to be predisposed to map moving agents identifying their leading and trailing ends in relation to their motion direction (i.e., to represent stimuli in terms of “head” and “tail”) [84], the motion pattern we employed might have impaired the differentiation of the left-right extremities of the stimuli. Another explanation for the lack of preference observed between AC and CA could be that chicks might have interpreted them as representing the same object when viewed from two different viewpoints. In fact, since A and C are symmetrical shapes, CA is the perfect mirrored image of AC (Figure 1c). Thus, if one imagines the AC stimulus as if it were painted on a translucent sheet, it should look identical to the CA stimulus to an observer standing behind the sheet (i.e., looking at it from the other side of the sheet). Future studies could test this possibility by using nonsymmetrical shapes to compose the stimuli.

Based on the results of Exp. 2 and 3, we could hypothesize that chicks learn only the identity of the elements located at the edges, without encoding any positional information (meaning that their representation of the imprinting object could be summarized as “the imprinting stimulus contains both the elements A and C”, which would support the preference for AC over AB and BC). However, the results of Exp. 1 allow us to exclude this possibility, since in this case both test stimuli (ABC and CAB) comprised all the three elements. Thus, the most parsimonious interpretation of the results is that chicks represent at least some positional information about the elements located at the stimulus edges, even though they do not discriminate between left and right margins. Chicks could learn that A and C should be located at the edge of the stimulus (ignoring their left-right orientation). Future studies should investigate whether chicks learn additional information from the imprinting stimulus (e.g., information about the central element, B).

Even though in the first two experiments chicks clearly revealed a significant preference for one of the two test stimuli, that preference was not very large (e.g., average proportion of preference of about 0.58 in the first experiment). This is consistent with what often reported in the literature on spontaneous social responses and filial imprinting [42,57,85,86,87], probably due to the spontaneous nature of this learning task. Moreover, in the natural environment a multitude of visual, acoustical and olfactory features can be used to discriminate the familiar imprinting object from an unfamiliar one. On the contrary, here, the two stimuli can be distinguished only by the ordering of their elements, making the difference between them harder to detect.

A most interesting result of the current study is chicks’ tendency to privilege information on the elements located at the edges of the imprinting stimulus. To the best of our knowledge, this is the first study to find this effect for spatial configurations of simultaneously presented elements, either in humans or nonhuman animals. Similar results had been previously reported for nonhuman primates with temporal acoustical sequences [35,43], in line with the importance of marginal elements in syntax acquisition. This mechanism seemed to be shared at least between primates, suggesting that it did not evolve specifically to support language, but it could rather reflect general functional constraints for the processing of temporal auditory sequences, then co-opted by linguistic and communicative functions. Studies with operant training in pigeons seemed to suggest that similar mechanisms might extend also to avian species and to temporally-organized sequences of visual elements (that have to be selected in a given sequence by the animals) [74,75,76] (a different interpretation is described in a previous paper [88]). However, until now, no evidence had ever been reported of an edge-advantage on avian species using unsupervised learning paradigms comparable to those employed with human and nonhuman primates [35,43]. Moreover, this represents also the first evidence of an edge advantage for purely spatial configurations, without temporal components defining their structure. This is particularly relevant, because it may indicate a higher level of generality of the underlying mechanism than originally thought, not only in terms of the phylogenetic distance between species, but also extending it beyond the acoustic modality, to the visual processing of patterns articulated over space, rather than over time. However, existing evidence suggests that general learning mechanisms might be modulated by factors like the sensory modality [7], as well as the distribution of regularities over space or over time and the specific learning task (e.g., imprinting vs. associative learning), making it crucial to experimentally verify assumptions on the presence of general underlying mechanisms.

## 7. Conclusions

To conclude, this work represents the first step in the investigation of unsupervised learning of spatial configurations of visual elements that differ only in shape, in a nonvocal learning model characterized by precocial development and a wide phylogenetic distance from the human species. These are all features that, in the most recent literature, are considered important to the understanding of the evolutionary history of the mechanisms underlying the development of communicative adaptations, such as human language and birdsong.

## Figures and Tables

**Figure 1 animals-08-00135-f001:**
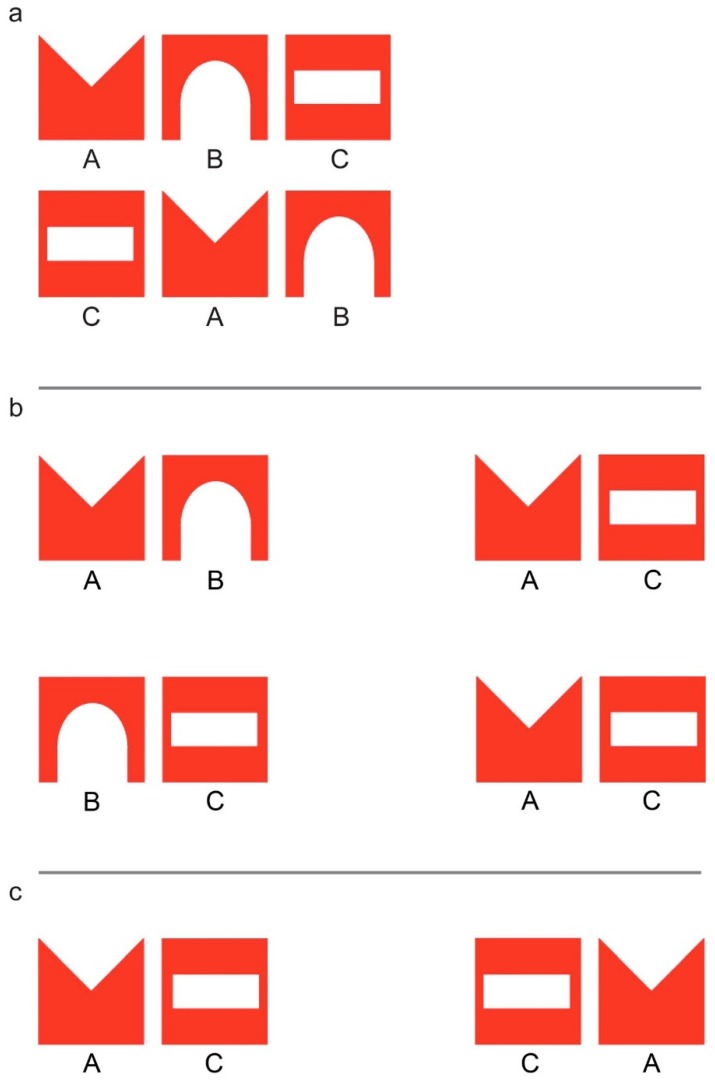
The triplets used as imprinting and test stimuli in Experiment (Exp.) 1, ABC and CAB. The triplet ABC was used as imprinting stimulus in Exp. 2 and 3 (**a**); the two pairs of fragments used as test stimuli in Exp. 2, AB vs. AC and BC vs. AC (**b**); the two fragments used as test stimuli in Exp. 3, AC vs. CA (**c**).

**Figure 2 animals-08-00135-f002:**
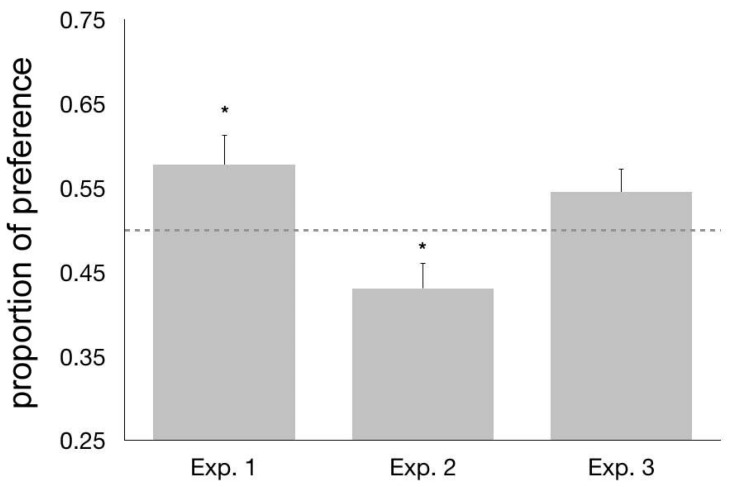
From left to right, the columns represent the average values of the preference (proportion of distance walked) for the familiar imprinting triplet (Exp. 1), for the bigram (AB or BC) that respects the between-elements adjacency-relationships (Exp. 2) and for the AC bigram (Exp. 3). Error bars represent the standard error of the mean (s.e.m.). The dotted line indicates the chance level (0.5), and * represent significant departures from chance (*p* < 0.05).

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
