# Peer review of "Spontaneous Learning of Visual Structures in Domestic Chicks"

_animals, 2018, doi:10.3390/ani8080135_

Round 1

Reviewer 1 Report

1)      The authors present the means and measures of variation in the text and in figure 1. Given that the presentation in the text is much shorter and can be clearly understood, this makes figure 1 superfluous. However, for people that are more visually guided, the figure would be all right

2)      I like the elaborate explanations of the reasons and procedures of the three separate experiments, It makes the experiments very clear.

3)      Although it is clear in experiment 1 that there is a significant preference to walk towards the familiar imprinting stimulus, the effect is not very large (average ratio of preference, which in fact is not a ratio but a proportion (line 259), is between 0.55 and 0.6), which is also the case for the proportions in experiment 2. The absence of a larger effect should be discussed. The test duration was 20 minutes ( line 251) which might have been too long to get a proper response, i.e. the chicks may have become frustrated in not being able to reach the imprinting object and then perhaps try to reach the other object. I think the magnitude of the response should be (shortly) discussed, and if possible describe the dynamics of running in the tests.

4)      When removed from an imprinting stimulus, chicks try to reach the stimulus to increase their ‘comfort’ and usually utter distress calls (and comfort calls when they reach the stimulus), is there any additional information, that can link preference to recognition, as preference is mostly interpreted as recognition throughout the text? Is there any information on distress calls in relation to the direction of movement? Is there any information on how the stimulus object compares to more natural objects, in other words, what do chicks gain from running to the imprinting stimulus?

5)      Section 2.3: Test procedure: it is not described whether the stimuli alternated between the two sides of the apparatus, in that case there may have been side biases that either caused te effect or reduced the effect of the stimuli in the three experiments. Please elaborate.

The reasoning in experiment three is not completely clear to me. When one presents an imprinting object that consists of two symmetrical object or pictograms (AC in the experiment), and one mirrors those, it in fact represents the back side of the stimulus. To me presenting the two simultaneously would mean that on one side the imprinting stimulus faces the chicks (original), whereas on the other side the imprinting stimulus faces away (mirrored condition), which may in certain chicks induce a follow response. So instead of testing whether chicks encode only the identity of the marks at the edges of the imprinting stimulus or also between left and right, one could argue that they tested whether the chicks had a mental representation of the back side of the stimulus, which may explain the lack of an effect in experiment 3. Although the authors discuss this in lines 467 to 472, they may elaborate a bit more on the functional significance of this.

Author Response

We are grateful to the Reviewer for Her/His comments and feedback. Below each comment we have reported our response to the point raised by the Reviewer and how we have modified the MS based on the Reviewer suggestions (the Reviewer’s comments are reported in grey here below).

1)      The authors present the means and measures of variation in the text and in figure 1. Given that the presentation in the text is much shorter and can be clearly understood, this makes figure 1 superfluous. However, for people that are more visually guided, the figure would be all right.

We are aware that the information presented in Fig. 1 is partially redundant, since the structure of the stimuli can be understood also from reading the text (same is true for Fig. 2 that reports the results). However, we agree with the Reviewer that the presence of the figure makes the reading of the paper more intuitive and might be appreciated by readers. We leave it to the Editor to decide whether Fig. 1 should be removed or should remain, based on the Journal policy.

2)      I like the elaborate explanations of the reasons and procedures of the three separate experiments, It makes the experiments very clear.

We thank the Reviewer for Her/His kind comments.

3)      Although it is clear in experiment 1 that there is a significant preference to walk towards the familiar imprinting stimulus, the effect is not very large (average ratio of preference, which in fact is not a ratio but a proportion (line 259), is between 0.55 and 0.6), which is also the case for the proportions in experiment 2. The absence of a larger effect should be discussed. The test duration was 20 minutes (line 251) which might have been too long to get a proper response, i.e. the chicks may have become frustrated in not being able to reach the imprinting object and then perhaps try to reach the other object. I think the magnitude of the response should be (shortly) discussed, and if possible describe the dynamics of running in the tests.

Indeed, as correctly pointed out by the Reviewer, the effects that we have observed here are not very large. This is not uncommon in research that investigates spontaneous social responses and  imprinting object recognition. Indeed, preference scores comprised between 0.55 and 0.60 have been reported by multiple studies of this kind in chicks: e.g., Vallortigara and Regolin (2006, Curr Biol); Vallortigara et al. (2005, PLoS Biol); Versace et al. (2017, Anim Cogn, for the recognition of the visual stimuli); Versace et al. (2006, Proceedings of The Evolution of Language); Vallortigara and Andrew (1994, Behav Proc). Thus, the results we obtained in the current study are in line with the previous literature in this animal model, including studies on imprinting learning and more specifically on sequence and rule learning. This is likely to reflect the spontaneous nature of the learning task and of the tests employed to study these phenomena, and the fact that our design relied on a focused manipulation of isolated features of the stimuli. In the natural environment, a multitude of visual, acoustical and olfactory features can be used to discriminate the familiar imprinting object from an unfamiliar one, whereas here (as in the studies cited above), the two stimuli can be distinguished only by the internal ordering of elements. All other visual features (shapes, colour, movement, size) are in fact identical between the two stimuli. Moreover, since imprinting learning is by definition a form of unconstrained learning by exposure, during the learning phase the animals are not shaped to pay attention to specific aspects of the stimuli (such as the ordering of the internal elements). Inevitably, some of the individuals will pay attention and encode only task-irrelevant elements of the stimuli (e.g. colour/size), weakening the effect at the group level. We have now briefly discussed this point in the General Discussion.

However, regarding the specific proposal of the Reviewer, we can exclude the duration of the test as a factor in determining the relatively small effect size observed here. In fact, if we consider only the very first minutes of the test, we do not observe a stronger preference for the familiar imprinting object than what we found analysing the entire duration of the test. E.g., in Exp. 1 the average proportion of preference for the imprinting stimulus during the first 5 minutes of the test is of 0.56, which is in line with the results for the whole test duration (0.57). The same is true for Exp. 2 with an average score of 0.43 for the first 5 minutes, a value identical to that observed for the whole test duration. To increase the possibility to compare our results with those of other experiments, though, we prefer to analyze the full 20 minutes of the test, as previously done in the literature in the field.

On a minor note, we have substituted the term “ratio” with the term “proportion” throughout the text, as suggested by the Reviewer.

4)      When removed from an imprinting stimulus, chicks try to reach the stimulus to increase their ‘comfort’ and usually utter distress calls (and comfort calls when they reach the stimulus), is there any additional information, that can link preference to recognition, as preference is mostly interpreted as recognition throughout the text? Is there any information on distress calls in relation to the direction of movement? Is there any information on how the stimulus object compares to more natural objects, in other words, what do chicks gain from running to the imprinting stimulus?

We did not analyse the calls emitted by the chicks or other behavioural measures except for the approach behaviour towards the stimuli. As correctly stated by the Reviewer, approaching the imprinting stimulus is the main measure used to determine successful imprinting and recognition of the imprinting object. Preferential approach has been the standard measure used to represent recognition of the imprinting stimulus in a large number of studies, from different labs, including the seminal works of Prof. Gabriel Horn and his collaborators that employed a very similar setting to that used in our study (based on the automated counting of approach attempts inside a running wheel) (e.g., see reviews in Horn, 2004, Nat Rev Neurosci; Bolhuis and Honey, 1998, Trends Neurosci; McCabe, 2013, WIREs Cogn Sci). Moreover, because we simultaneously present two stimuli at test, measures such as distress calls would be difficult to interpret as in reference to one specific stimulus.

Indeed, since the adaptive value of imprinting is to ensure that the chicks will maintain contact with the mother hen and the rest of the brood, preferential approach of the imprinting object with respect to other objects is the key defining feature of imprinting.

It is also widely recognised that, in the absence of more natural objects, the process of imprinting occurs effectively also towards artificial stimuli that bear very little resemblance with natural objects, as it is the case in our study (see the references cited above).

5)      Section 2.3: Test procedure: it is not described whether the stimuli alternated between the two sides of the apparatus, in that case there may have been side biases that either caused te effect or reduced the effect of the stimuli in the three experiments. Please elaborate.

We thank the Reviewer for pointing out this potential element of confusion. In the current study we did not alternate the position of the stimuli between the two sides of the apparatus within a test trial. However, the side of the presentation of the familiar and the unfamiliar bigram was counterbalanced between subjects, a standard practice in this field of research. In this way we ruled out that any side bias due to a preference for one side of the apparatus or any other environmental asymmetry, might have caused the effects observed here. This has been now explicitly clarified at lines 249-251 (line numbers refer to MS version without track changes).

6) The reasoning in experiment three is not completely clear to me. When one presents an imprinting object that consists of two symmetrical object or pictograms (AC in the experiment), and one mirrors those, it in fact represents the back side of the stimulus. To me presenting the two simultaneously would mean that on one side the imprinting stimulus faces the chicks (original), whereas on the other side the imprinting stimulus faces away (mirrored condition), which may in certain chicks induce a follow response. So instead of testing whether chicks encode only the identity of the marks at the edges of the imprinting stimulus or also between left and right, one could argue that they tested whether the chicks had a mental representation of the back side of the stimulus, which may explain the lack of an effect in experiment 3. Although the authors discuss this in lines 467 to 472, they may elaborate a bit more on the functional significance of this.

We thank the reviewer for providing this interesting potential alternative explanation for the results of Exp. 3. We now have incorporated this element in the General Discussion at lines 486 and following.

Reviewer 2 Report

 This is an interesting paper, but as it stands very hard work to read. There is much of jargon used from research on machine learning that needs to be carefully translated in order to be  of interest to the readership of Animals.

There is much repetition too, which must be omitted and perhaps more succinct statements made to make it easier without loosing scientific respectability.

However the idea of applying "statistical learning" to non language  uses and not in

communication, is interesting.This I understand, from reading, is either or both "a mechanism designed to detect structures inherently  in  the environment which are important in cognitive development" and or "a tool for developing practical  algorythms for estimating multi dimensional functions."   In other words the measurment of the acquistion of knowledge about the environment and combining this with experiences?

In these experiments, the issue is "silent learning" ( i.e. learning without reinforcement), which they assume is  characteristic of imprinting or early learning, but I would like more references here to convince me of this since "imprinting" is usually defined just as "early learning"?

The paper however, needs rewriting using simple terminology rather than that  complex terms applicable to machine learning, or at least, the terms should be clearly and simply defined.

It should be shortened by around 4 pages . If this is done, and the introduction includes why these experiments are of interest to Animal readers, in concise and simple terms, then this will be a valuable contribution. As it stands, it is not.

Detailed comments follow.

99-100 What is Domain general? This has been used to indicate  "generalisation", for example, an interest in novel stimuli. No novel stimuli used here, and are you sure that the chicks would not show much interest in a novel stimulus, did you try it? .... Define your terms simply.

117-119 Repeat statements, These occur several times and to clarify they MUST be omitted. The points should be  made with clarity, simply and succintly.

168 Rewrite, shorten reduce jargon`, eg "triplet state".

171-172, well yes but why NOT?

258 "collapse", you mean joined? or ignored?

310-313 Strings are  an encoded linear  property?

325-26 repeat

364 Not very clear,  why should edges be spatial  and not temporal on this moving target?

435-450 Rewrite and shorten. 

Author Response

We have now revised the manuscript taking into account the feedback provided by the Reviewer, with particular attention to improving the readability of the manuscript and avoiding repetitions, and we believe the paper is now stronger because of the feedback. Here below we report our answers to the specific points raised by the Reviewer (the Reviewer’s comments are reported in grey here below).

This is an interesting paper, but as it stands very hard work to read. There is much of jargon used from research on machine learning that needs to be carefully translated in order to be  of interest to the readership of Animals. There is much repetition too, which must be omitted and perhaps more succinct statements made to make it easier without loosing scientific respectability. 

We have revised the manuscript both in the introduction and in the discussion in order to improve readability. We would like to point out, however, that the terminology we use here is not related to machine learning, as the Reviewer seems to believe: the concepts and the terminology used here are standard elements in the field of animal cognition (Abe and Watanabe, 2011 Nat Neurosci; Chen and Ten Cate, 2015 Behav Processes; Chen et al., 2015 Anim Cogn; Comins and Gentner, 2013 Cognition; de la Mora and Toro, 2013 Cognition; Endress et al., 2010 Anim Cogn; Fehér et al., 2017 Philos Trans; Gentner et al., 2006 Nature; Goujon and Fagot, 2013 Behav Brain Res; Grainger et al., 2012 Science; Hauser and Glynn, 2009 J Comp Psychol; Hauser et al., 2001 Cognition; Martinho and Kacelnik, 2016 Science;  Menyhart et al., 2015 Front Psychol;  Murphy et al., 2008 Science; Neiworth et al., 2017 J Comp Psychol; Saffran et al., 2008 Cognition; Scarf and Colombo, 2008 Brain Res Bull; Scarf and Colombo, 2010 J Exp Psychol; Spierings and Ten Cate, 2016 PNAS; Sonnweber et al., 2015 Anim Cogn; Stobbe et al. 2012 Philos Trans; Straub and Terrace 1981 Anim Learn Behav; Takahasi et al., 2010 Ethology; ten Cate and Okanoya, 2012 Philos Trans; Terrace, 1987 Nature; Toro and Trobalón, 2005 Percept Psychophys; van Heijningen et al., 2016 Anim Cogn; Wilson et al., 2013 J Neurosci), which is certainly within the interest of the readers of “Animals”. In particular, being part of a special issue on “Animal Communication”, the paper must necessarily refer to concepts and terminology related to animals’ cognitive capacities implied in communication and we are confident that in this revision we have clarified less than clear sentences.

However the idea of applying "statistical learning" to non language uses and not in communication, is interesting. This I understand, from reading, is either or both "a mechanism designed to detect structures inherently in the environment which are important in cognitive development" and or "a tool for developing practical  algorythms for estimating multi dimensional functions."   In other words the measurment of the acquistion of knowledge about the environment and combining this with experiences? 

We thank the reviewer for his kind comment. Statistical learning in this context is defined as a mechanisms that allows “acquisition of structured information and the detection of regularities embedded in the sensory input” (see lines 77-79, line numbers refer to MS version without track changes), so it certainly reflects the acquisition of knowledge from experience with the environment, as stated by the Reviewer.

In these experiments, the issue is "silent learning" ( i.e. learning without reinforcement), which they assume is  characteristic of imprinting or early learning, but I would like more references here to convince me of this since "imprinting" is usually defined just as "early learning”?

Imprinting cannot be simply defined as early learning, there are many forms of early learning that are different from imprinting. From Lorenz onward, filial imprinting has been specifically defined as a specialised form of unrewarded learning (or learning by exposure/observation) that restricts affiliative response of nidifugous hatchlings to a salient object they are exposed to, without any specific reinforcement being provided to the animal (e.g., Lorenz, 1935, J Ornithol; Bateson, 1979, Anim Behav; Bateson, 1979, Anim Learn Behav; Bolhuis and Honey, 1998, Trends Neurosci; McCabe, 2013, WIREs Cogn Sci; Horn, 2004, Nat Rev Neurosci) in a specific time window.

The paper however, needs rewriting using simple terminology rather than that complex terms applicable to machine learning, or at least, the terms should be clearly and simply defined. It should be shortened by around 4 pages . If this is done, and the introduction includes why these experiments are of interest to Animal readers, in concise and simple terms, then this will be a valuable contribution. As it stands, it is not. 

See our previous answer above.

Detailed comments follow.

99-100 What is Domain general? This has been used to indicate "generalisation", for example, an interest in novel stimuli. No novel stimuli used here, and are you sure that the chicks would not show much interest in a novel stimulus, did you try it? .... Define your terms simply.

The term “domain general” does not refer to generalisation to novel stimuli. Rather, this standard term in the field of cognitive science refers to processes or mechanisms that operate across different cognitive domains (e.g. mechanisms that work the same way with spatial or social stimuli). Statistical learning is often considered to be a domain-general mechanism because it operates across multiple domains including language, music, and recognition of visual objects. To increase the manuscript readability, we have now avoided to explicitly refer to the concept of “domain generality”.

117-119 Repeat statements, These occur several times and to clarify they MUST be omitted. The points should be  made with clarity, simply and succintly.

168 Rewrite, shorten reduce jargon`, eg "triplet state”.

We have overall revised and shortened the MS according to the reviewer’s request. However, the expression “triplet state” has never been used in the paper.

171-172, well yes but why NOT?

We could not understand this comment, so if the Reviewer meant to refer to a specific issue with our manuscript which we did not address already with our changes, we ask him/her to clarify it.

258 "collapse", you mean joined? or ignored?

The data were analysed together, as it is customary and required when no statistically significant difference is found between different groups of subjects. This has been clarified at lines 308-311 and 379.

310-313 Strings are  an encoded linear  property?

We originally defined “strings” at the beginning of the text (in the Short Summary), as “configurations composed of multiple visual elements presented simultaneously side-by-side”. However, we have avoided this term in the revised version of the manuscript.

325-26 repeat

We have revised the MS to avoid repetitions.

364 Not very clear,  why should edges be spatial  and not temporal on this moving target?

This is because all the elements of the stimulus are always simultaneously visible, they do not appear one by one on the screen. Thus the structure of the stimulus is defined by their relative spatial positions and not by their temporal order of appearance. This has been clarified at the beginning of the text, lines 168-171. 

435-450 Rewrite and shorten.

See above, we have revised the MS for clarity and to avoid repetitions. 

Reviewer 3 Report

Introduction: Overall, the introduction could be strengthened by removing some redundant material, thereby shortening some of the paragraphs. This would make it easier for the reader. P2 L51: Consider changing to a general mechanism, or delete "a". P2 L76: Change For examples to For example P3 L105: Refrain from using the term "proven", as research lends support/evidence, but doesn't prove. Materials and Methods/Results: In the home cages, during imprinting, was there a visual barrier between chicks? Authors point out the reduction of animals used in these experiments by excluding males. The number of females used varies in each experiment. Please explain how the researchers got to that number in each experiment. P5 L212: Remove one set of "in a". P8 L265: change run to ran P9 L316: remove one set of "the most" General Discussion: Again, some repeating of information, this section can be streamlined like the introduction.

Author Response

We thank the Reviewer for Her/His comments (which are reported in grey here below). We have implemented the corrections that he suggested, modifying the MS accordingly and we have provided here below our detailed answers to each specific comment. In particular we have shortened the Introduction and the General Discussion, as suggested by the Reviewer and we believe that the paper is now more fluent to read.

Introduction: Overall, the introduction could be strengthened by removing some redundant material, thereby shortening some of the paragraphs. This would make it easier for the reader.

We have now shortened the Introduction by removing redundant material, as suggested.

P2 L51: Consider changing to a general mechanism, or delete "a".

P2 L76: Change For examples to For example

P3 L105: Refrain from using the term "proven", as research lends support/evidence, but doesn't prove.

These changes have been made.

Materials and Methods/Results: In the home cages, during imprinting, was there a visual barrier between chicks?

Chicks could not see each other, since they were caged individually in black plastic cages that separated individuals visually. This was done in order to prevent visual imprinting on conspecifics, which could have impaired learning about the artificial imprinting object visible on the screen that was present on the front wall of the cage. This has now been clarified at lines 196-199 (line numbers refer to MS version without track changes).

Authors point out the reduction of animals used in these experiments by excluding males. The number of females used varies in each experiment. Please explain how the researchers got to that number in each experiment.

The number of chicks tested varies slightly from experiment to experiment, because we included all the female chicks that provided data within each hatching batch we tested. Our aim for the sample size in this study was to reach about 60 chicks for experiment, which we considered, based on previous literature and our experience, the minimum sample size to observe reliable effects with this paradigm. Exp. 3 has a slightly bigger sample size than the first two, but we believe that this is important since in this experiment we do not observe any significant departure from chance level and a bigger sample size allows us to be more confident in interpreting this result (i.e. decreases the chances that an effect would be there and could be observable if we just had enough statistical power to detect it).

P5 L212: Remove one set of "in a".

P8 L265: change run to ran

P9 L316: remove one set of "the most"

These changes have been done, thanks.

General Discussion: Again, some repeating of information, this section can be streamlined like the introduction. 

As for the Introduction, we removed repeating information in order to shorten this section.

Round 2

Reviewer 2 Report

This is improved, but still could be made clearer for those not accustomed to the jargon, which will be the readers of Animals which, as I understand it, is a multi disciplinary journal committed to further knowledge concerning in particular animal welfare.

The authors have certainly rapped me over the knuckles concerning "statistical learning", but however many cognitive ethologists have used this term, it is in origin from machine learning, and one wonders how it may make things clearer for sentient beings who we are already aware that  learn takes into account environmental variables as well an emotion  ( motivation) to learn.  Because others do it, does not mean that everyone should, particularly if it is not clear.

But nevertheless, I would suggest that if they re-revise it one more time, shortening another page or so, that this ms will be a valuable contribution to the multi-disciplinary field of animal cognition , and chick learning... although the relevance to other chicks remains obscure since these chicks were raised in black plastic, not exactly the normal experience!... May be there are some ethical concerns here but I will let this pass.

I have written some detailed comments but they have not been forwarded and alas  I do not have a copy, however they should look at  lines:- 100, 282-283 grammar, 626 spelling, 800 you mean chicks? 812 in the literature?

820-821 rewrite shorten 869-882 not clear about this, how can they look from back to front?, clarify the point you are making in one sentence.

Author Response

Based on this further feedback provided by the Reviewer we have additionally revised the manuscript. In particular we have shortened it, we have clarified the more  complex passages and we have tried to avoid the use of “technical” terminology from the field of cognitive sciences, as much as we could without losing the meaning of the text.

As regards the ethical aspect of the study, we wish to point out that all the procedures were approved by the competent authorities, in line also with European guidelines (as specified in the manuscript). In particular, temporarily raising chicks for few days in the presence of an artificial imprinting object (while keeping them in auditory and olfactory contact with the brood-mates) is a standard practice that is not distressful to the animals and does not constitute a form of social deprivation. This is also apparent by the fact that, while they are in their cages with the imprinting object chicks behave normally and do not emit distress calls, but rather soft calls that are an index of contentment. In fact, chicks rapidly imprint on the visual stimulus they are exposed to, directing their social affiliative responses to it until they are rejoined with conspecifics at the end of the experiment.